# First Glance of Molecular Profile of Atypical Cellular Angiofibroma/Cellular Angiofibroma with Sarcomatous Transformation by Next Generation Sequencing

**DOI:** 10.3390/diagnostics10010035

**Published:** 2020-01-09

**Authors:** Yi-Che Chang Chien, Attila Mokánszki, Hsuan-Ying Huang, Raimundo Geronimo Silva, Chien-Chin Chen, Lívia Beke, Anikó Mónus, Gábor Méhes

**Affiliations:** 1Department of Pathology, University of Debrecen, Clinical Center, 4032 Debrecen, Hungary; mokanszki.attila@med.unideb.hu (A.M.); livia.beke@med.unideb.hu (L.B.); monusne.aniko@med.unideb.hu (A.M.); gabor.mehes@med.unideb.hu (G.M.); 2Department of Pathology, Kaohsiung Chang Gung Memorial Hospital and Chang Gung University College of Medicine, Kaohsiung 83301, Taiwan; a120600310@yahoo.com; 3Department of Pathology, Federal University of Piaui, Teresina 64049-550, Brazil; gerjrpi@gmail.com; 4Department of Pathology, Ditmanson Medical Foundation Chia-Yi Christian Hospital, Chiayi 600, Taiwan; hlmarkc@gmail.com; 5Department of Cosmetic Science, Chia Nan University of Pharmacy and Science, Tainan 717, Taiwan

**Keywords:** atypical angiofibroma, angiofibroma with sarcomatous transformation, p16, p53, next generation sequencing

## Abstract

Cellular angiofibroma is a rare benign mesenchymal neoplasm most commonly occurring in the vulvovaginal region in women and the inguinoscrotal region in men with specific genetic deletion involved in the *RB1* gene in chromosome 13q14 region. Atypical cellular angiofibroma and cellular angiofibroma with sarcomatous transformation are recently described variants showing worrisome morphological features and strong, diffuse p16 expression. Nevertheless, the molecular profile of these tumor entities is largely unknown. We carried out a next generation sequencing (NGS) study from six cases of atypical cellular angiofibroma and cellular angiofibroma with sarcomatous transformation. We were able to identify oncogenic *TP53* gene mutations (33%) which may contribute to pathogenesis also resulting in p16 overexpression. In addition, *RB1* gene alterations generally present were identified. Since it is a recently described and rare entity, the whole molecular signaling pathway is still largely obscured and the analysis of larger cohorts is needed to elucidate this issue.

## 1. Introduction

Cellular angiofibroma (CA) is a benign mesenchymal neoplasm most commonly occurring in the vulvovaginal region in women and in the inguinoscrotal region in men [1]. The tumor is characterized by a spindle cell proliferation intermixed with hyalinized small-to-medium-sized blood vessels. It was first described in 1997 [2], and more recently, a specific genetic deletion involving the *RB1* gene at chromosome region 13q14 was documented [3,4], indicating a close relationship with spindle cell lipoma, mammary-type myofibroblastoma, and to a certain degree, atypical spindle cell lipomatous tumor [5]. Atypical cellular angiofibroma (ACA) and cellular angiofibroma with sarcomatous transformation (CAS) are rare entities of CA showing atypical morphology features. The former usually has nuclear atypia and the latter presents prominent sarcomatous overgrowth with features of well-differentiated liposarcoma (WDLPS) without *MDM2* gene amplification, pleomorphic liposarcoma (PLPS) or undifferentiated pleomorphic sarcoma (UPS) [6]. Interestingly, both ACA and CAS show relatively good prognosis and no tumor associated mortality has yet been documented [6]. However, the biological significance of ACA/CAS remains uncertain; to the best of our knowledge, apart from the deletion of *RB1* gene, no detailed molecular data are published in the literature. This triggered our interest to learn whether there may be differences regarding the genetic profiles between CA and ACA/CAS. Furthermore, it is known that ACA/CAS also show strong and diffuse p16 expression in atypical cells and the sarcomatous component, suggesting an underlying molecular mechanism involved in the oncogenesis [6]. Hence, we carried out a 67 gene next generation sequencing (NGS) study from six cases of ACA/CAS and two cases of CA (served as control group) to provide a deeper molecular insight into both groups.

## 2. Method and Materials

Six cases of ACA/CAS were collected between 2014 and 2019 (three cases from personal extramural consult archive of one author (HYH), Taiwan; one case from University of Debrecen Clinical Center, Debrecen, Hungary; one case from Chia-Yi Christian Hospital, Chia-Yi, Taiwan; and one from Lapac Patologi Cirurgia Molecular, Teresina, Brazil). Two cases of CA (Debrecen, Hungary) were used as the control group. All protocols were approved by the authors’ respective Institutional Review Board for human subjects (IRB reference number: 60355/2016/EKU). Hematoxylin and eosin stained sections of all cases were reviewed by the same pathology consultant. The results of immunohistochemical studies were provided by the referring pathologists. Additional staining for p16INK4a (dilution: 1:2, MTM Laboratories, Heidelberg, Germany), p53 (dilution 1:1200, Immunotech, Prague, Czech Republic), MDM2 (dilution 1:50, Calbiochem, Merck, Kenilworth, NJ, USA), and CDK4 (dilution: 1:800, Biosource, Thermo Fisher Scientific, Waltham, MA, USA) was also carried out.

Fluorescence in situ hybridization (FISH) was performed on 5 µm thick sections of formaldehyde fixed and paraffin embedded (FFPE) samples with XL RB1/DLEU/LAMP deletion probe (MetaSystems, Altlussheim, Germany) according to the manufacturer’s protocol. Deparaffinized sections (Q Path Safesolv, VWR, Debrecen, Hungary) were pretreated with Pretreatment Buffer followed by proteolytic digestion using Protease Solution (MetaSystems, Altlussheim, Germany). Slide and probe codenaturation was carried out at 75 °C for 5 min and hybridization was provided at 37 °C in a moist chamber for 16–18 h (StatSpin ThermoBrite, Abbott Molecular, Des Plaines, IL, USA). Post-hybridization washes were performed with 2× saline-sodium citrate (SSC) for 5 min. The slides were then washed with 0.4× SSC at 72 °C for 2 min and 2× SSC/0.05% Tween 20 for 2 min. After washing, the nuclei were counterstained with 4’-6’ diamidino-2-phenylindole (DAPI, MetaSystems, Altlussheim, Germany). Scoring was performed using Zeiss Axio Imager Z2 (Carl Zeiss, Cambridge, UK) fluorescence microscope; the images were captured and analyzed by ISIS software (MetaSystems, Altlussheim, Germany).

Genomic DNA was extracted from formaldehyde fixed and paraffin embedded (FFPE) tissues using the QIAamp DNA FFPE Tissue Kit (Qiagen, Hilden, Germany) according to the manufacturer’s protocol. DNA concentration was measured in Qubit dsDNA HS Assay Kit using a Qubit 4.0 Fluorometer (Thermo Fisher Scientific, Waltham, MA, USA). Amplifiable genomes were calculated using Archer PreSeq DNA Calculator according to Archer PreSeq DNA QC Assay Protocol (Archer DX, Boulder, CO, USA).

After fragmentation of the genomic DNA, libraries were created using Archer VariantPlex Solid Tumor Kit (Archer DX, Boulder, CO, USA). The final libraries were quantified using KAPA Universal Library Quantification Kit (Kapa Biosystems, Roche, Basel, Switzerland).

The libraries were diluted to a final concentration of 4  nM and pooled by equal molarity. To sequence using the MiSeq System (MiSeq Reagent kit v3 2 × 300 cycles), all libraries were denatured by adding 0.2 nM NaOH and diluted to 40 pM with hybridization buffer from Illumina (San Diego, CA, USA). The final loading concentration was 14 pM libraries and 5% PhiX. Sequencing was conducted according to the MiSeq instruction manual. The data were analyzed with the Archer analysis software (Version 6.2.; Archer DX, Boulder, CO, USA) for the presence of single-nucleotide variants (SNVs) and indels. A reliable variant detection required a coverage of >250 reads. For alignment, the human reference genome GRCh37 (equivalent UCSC version hg19) was built. The results of NGS was cross-checked using COSMIC (Catalogue of Somatic Mutations in Cancer) and cBioPortal databases. We used gnomAD v.2.1.1 population database to compare the significance of each gene’s alterations which is included the Archer NGS analysis system. We included single nucleotide changes that occurred in the canonical transcript that were found at a frequency of >0.001%, passed all filters, and at sites with a median depth ≥1.

## 3. Result

The microscopical analysis of all six ACA/CAS cases showed typical cellular angiofibroma areas composed of uniform, bland spindle-shaped cells haphazardly arranged in a collagenous stroma with numerous thick-walled and hyalinized blood vessels. One case of ACA with scattered atypical cells possessed hyperchromatic nuclei (Figure 1) and the other five cases of CAS had two to three PLPS- and UPS-like areas (Figure 2A). Neither atypical mitoses nor tumor necrosis was found. Clinical features and the immunohistochemical (IHC) profile of the six ACA/CAS and the two CA cases are summarized in Table 1. The male and female ratio was 1:2 and all cases were located within the pelvic-perineal region. The tumor size varied from 4 to 9 cm. Immunohistochemically, all ACA/CAS cases showed positivity for CD34, estrogen/progesterone receptor, and also p16 (Figure 2B). None of the cases showed positivity for MDM2 and CDK4 (the complete results of immunohistochemical staining is provided in Appendix A). Both control cases were negative for p16 by immunohistochemistry (pictures not shown).

The results of the 67 solid tumor gene panel NGS are summarized in Table 2. The cutoff variant allele frequency (VAF) was determined at 2%. Large insertion/deletions (>50 base pairs) and complex mutations were not detected by this approach.

Based on our NGS results, two out of six ACA/CAS cases (case 2 and 6) carried high levels of *TP53* pathogenic gene mutations (c.626_627del; p.Arg209LysfsTer6; VAF = 39.2% and c.488A>G; Tyr163Cys; VAF = 63.6%) and p53 immunohistochemical staining for both cases showed strong intranuclear positivity within the sarcomatous component as well (Figure 2C).

NGS revealed further gene alterations as well. In case 1, two variants were detected in the *SMO* and *RHOA* genes which were classified as of uncertain significance according to the database. In case 2, four different variants were detected including one pathogenic *TP53* mutation (c.626_627del; p.Arg209LysfsTer6) and a benign *PTEN* variant. In case 3 and 5, two benign/uncertain *TP53* variants were detected at low VAF. In case 4, one pathogenic mutation was described (*HNF1A*, c.872dup; p.Gly292ArgfsTer25) at low VAF. In case 5, one pathogenic (*ALK*, c.3385G>A; p.Glu1129Lys) and one benign (*STK11*, c.1062C>G; p.Phe354Leu) variant were detected. In case 6, one pathogenic *NRAS* mutation was found (c.436G>A; p.Ala146Thr, VAF = 2.8%). No significant genetic variants could be demonstrated in any of the control CA cases.

Fluorescence in situ hybridization (FISH) examination showed a monoallelic deletion of *RB1* gene in all six ACA/CAS cases, similar to the control CA samples (Figure 2D, Appendix A).

## 4. Discussion

ACA/CAS was described as a separate new entity with a spectrum of distinctive morphological features covering cytological atypia and sarcomatous growth patterns. The diagnosis of ACA/CAS may be challenging particularly if the atypical/sarcomatous areas are predominant, emphasizing the importance of thorough sampling to represent typical CA areas. The presence of *RB1* gene deletion (e.g., by FISH analysis) was also found useful as a characteristic marker to differentiate from WDLPS, PLPS, and UPS, since the former has *MDM2* gene amplification and the latter two possess complex karyotypes.

It is documented that ACA/CAS often show either multifocal or diffuse p16 overexpression compared with either scattered or negative expression in CA [6], indicating additional molecular pathways crucial in ACA/CAS pathogenesis. The p16 protein, a product of *CDKN2A* (cyclin-dependent kinase inhibitor 2A, *p16^Ink4a^* and numerous other synonyms) gene is involved in cell cycle regulation and considered as a tumor suppressor protein with a potential role in cell senescence, apoptosis, and angiogenesis. The downregulation and reduced p16 protein expression was reported in up to 50% of overall human malignancies [7,8]. Intriguingly, overexpression of p16 has also been described in several cancer types by diverse mechanisms. Briefly, p16 overexpression in tumors can be categorized as human papilloma virus (HPV)-associated or non-HPV-associated mechanisms. The former is due to p53 and *Rb* inhibition by the HPV E6 and E7 oncoproteins, respectively, due to the lack of negative feedback control (e.g., carcinoma of the uterine cervix) [9]. Alternatively, p16 overexpression may occur due to *RB1* and *TP53* gene mutations which further dysregulate the cell cycle signaling pathway, such as in serous endometrial carcinoma [10].

To elucidate the genetic background of this rare neoplasia we carried out a 67 gene panel NGS study covering the most important known oncogenic variants in solid malignancies. Following DNA isolation and successful sequencing from archived tissue samples, a small set of gene variants was identified in CAS cases but none in the CA or ACA cases, which might explain the higher mitotic activity and larger size of the former. As the most interesting finding, clinically significant *TP53* mutations were found in two out of five cases with CAS morphology which is in agreement with the sarcomatous transformation. The positive p53 IHC reaction in these two samples clearly supports deficient p53 functionality. Although case numbers were limited, *TP53* gene mutations were associated with larger tumor size and higher mitotic activity in our cohort, suggesting faster tumor growth rate in association with p53 dysfunction. The remaining four cases of ACA/CAS did not reveal any clinically relevant pathogenic mutations according to the COSMIC database. However, low frequency benign/undetermined gene variants might indicate subclonal changes indicating an enhanced genomic instability in ACA/CAS. We also double-checked our COSMIC results in the cBioPortal database to exclude the possibility of germline mutations.

Despite the sequence analysis of 67 genes, we still have an incomplete picture of the mechanisms potentially contributing to p16 overexpression besides *TP53* gene mutations. As such, CDK4 overexpression has been described in ACA in association with p16 upregulation [11]. Although we did not obtain results supporting this finding, it is clear that alternative biological processes influence both the morphology and the behavior of this recent entity. However, based on most recent clinical follow-ups, none of the six ACA/CAS cases evaluated here had tumor recurrence or metastasis implying that the atypical morphology, the sarcomatous transformation, and even TP53 mutations in cellular angiofibroma do not necessarily indicate aggressive clinical behavior.

In summary, this is the first study performing mutation profiling of ACA/CAS by NGS. In addition to sporadic and unclear gene variants, we were able to identify clinically relevant oncogenic *TP53* gene mutations in a significant part of CAS. Further, p53 deficiency may be involved in the tumorigenesis inducing p16 overexpression in selected cases. Since it is a recently described and rare entity, molecular signaling pathways of the ACA/CAS spectrum are still obscured and the analysis of larger sample cohorts is required.

## Figures and Tables

**Figure 1 diagnostics-10-00035-f001:**
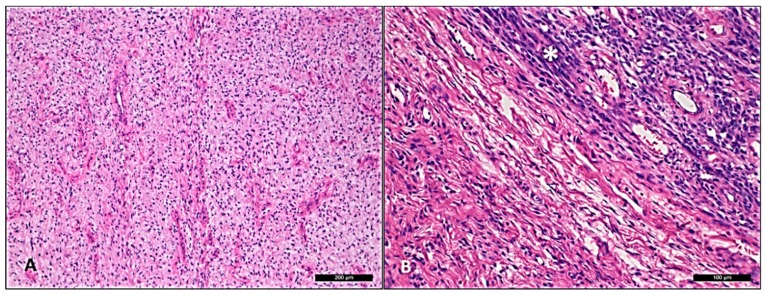
Atypical cellular angiofibroma with scattered atypical cells possessing hyperchromatic nuclei (case 5). (**A**) Area of usual cellular angiofibroma with bland spindle cell component with hyalinized vascular component in fibrotic stroma. (**B**) Abrupt transition to an area showing hypercellularity and moderate nuclear atypia labeled as “*”.

**Figure 2 diagnostics-10-00035-f002:**
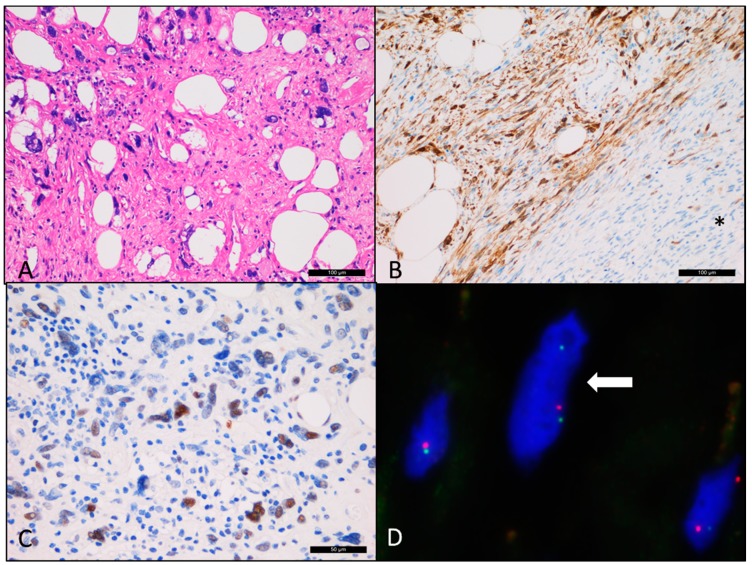
Cellular angiofibroma with sarcomatous transformation (case 6). (**A**) Sarcomatous area showing morphology similar to pleomorphic liposarcoma. (**B**) Diffuse p16 immunopositivity compared with the typical cellular angiofibroma regions labeled as “*” (negative). (**C**) The atypical cells showing positivity for p53. (**D**) Deletion of the *RB1* gene region (single red dot) detected by fluorescence in situ hybridization.

**Table 1 diagnostics-10-00035-t001:** Clinical data and immunohistochemical profile of six atypical angiofibroma/angiofibroma with sarcomatous transformation and two cellular angiofibroma control cases.

Case No.	Gender	Age	Location	Size (cm)	Depth	Atypical Areas	50/HPF	Atypical MF	Necrosis (%)	Infiltration	ER/PR	CD34	p16	p53	RB1 FISH	Follow-up
1	F	68	pelvis	4	deep	CAS with PLPS-like areas	2	no	0	circumscribed	1+	3+	3+	negative	positive	16 months
**2**	**M**	**36**	**hip**	**9**	**deep**	**CAS with UPS-like areas**	**8**	**no**	**0**	**mixed**	**1+**	**3+**	**3+**	**positive**	**positive**	**12 months**
3	M	73	inguinal	3.5	superficial	CAS with PLPS-like areas	0	no	0	circumscribed	3+	3+	2+	negative	positive	10 months
4	F	48	perineum	2.3	superficial	CAS with UPS-like areas	4	no	0	circumscribed	3+	3+	2+	negative	positive	12 months
5	F	25	vulva	4.3	deep	ACA	0	no	0	circumscribed	3+	1+	2+	negative	positive	8 months
**6**	**F**	**43**	**vulva**	**7**	**deep**	**CAS with PLPS-like areas**	**6**	**no**	**0**	**circumscribed**	**3+**	**3+**	**2+**	**positive**	**positive**	**6 months**
CA1	M	63	inguinal	7.5	deep		2	no	0	circumscribed	1+	2+	negative	negative	positive	12 months
CA2	F	52	vulva	2.2	superficial		0	no	0	circumscribed	1+	2+	negative	negative	positive	24 months

Bold letters represent the p53 positive cases. CA: cellular angiofibroma, ACA: atypical angiofibroma, CAS: angiofibroma with sarcomatous transformation, PLPS: pleomorphic liposarcoma, UPS: undifferentiated pleomorphic sarcoma, ER/PR: xxx, FISH: fluorescence in situ hybridization; 1+, 2+, and 3+ indicate mild, moderate, and strong positivity by immunohistochemical stain.

**Table 2 diagnostics-10-00035-t002:** Next generation sequencing result of six ACA/CAS cases.

Case No.	Gene	Mutation	Transcript IDs	VAF (%)	Significance	gnomAD Frequency (%)
1	*SMO*	c.743C>T; p.Thr248Ile	NM_005631.4	2.5	Uncertain	-
	*RHOA*	c.178A>G; p.Thr60Ala	NM_001313941.1	2.3	Uncertain	-
2	*PTEN*	c.511C>G; p.Leu171Val	NM_001304717.5	53	Heterozygous, benign	0.4
	***TP53***	**c.626_627del; p.Arg209LysfsTer6**	**NM_000546.5**	**39.2**	**Pathogenic**	**-**
	*H3F3A*	c.89C>T; p.Ala30Val	NM_002107.4	3.4	Uncertain	-
	*PIK3CA*	c.916T>C; p.Ser306Pro	NM_006218.2	3	Uncertain	-
3	*CDH1*	c.1174C>T; p.His392Tyr	NM_001317184.1	3.1	Uncertain	-
	*TP53*	c.3G>A; p.Met1	NM_001126118.1	2.8	Uncertain	-
	*APC*	c.6710G>A; p.Arg2237Gln	NM_000038.5	2.4	Uncertain	-
	*KIT*	c.148G>A; p.Val50Met	NM_000222.2	2	Benign	0.001
4	*FBXW7*	c.732T>G; p.Asp244Glu	NM_001013415.1	4.4	Uncertain	-
	*PIK3CA*	c.44T>G; p.Leu15Trp	NM_006218.2	3.4	Uncertain	-
	*HNF1A*	c.872dup; p.Gly292ArgfsTer25	NM_000545.7	2.1	Pathogenic	0.03
5	*STK11*	c.1062C>G; p.Phe354Leu	NM_000455.5	56	Heterozygous, benign	0.5
	*ALK*	c.3385G>A; p.Glu1129Lys	NM_004304.4	3	Pathogenic	-
	*EGFR*	c.2340G>A; p.Met780Ile	NM_001346897.1	2.7	Uncertain	-
	*MET*	c.3943G>A; p.Val1315Ile	NM_000245.2	2.4	Uncertain	-
	*TP53*	c.328C>T; p.Arg110Cys	NM_000546.5	2	Benign	0.002
6	***TP53***	**c.488A>G; Tyr163Cys**	**NM_000546.5**	**63.6**	**Pathogenic**	**-**
	*SMO*	c.1097C>T; p.Ser366Leu	NM_005631.4	4.6	Uncertain	-
	*JAK3*	c.2141C>T; p.Thr714Met	NM_000215.3	4.5	Uncertain	0.004
	*MYC*	c.1240G>A; p.Ala414Thr	NM_002467.4	3	Uncertain	-
	*NRAS*	c.436G>A; p.Ala146Thr	NM_002524.4	2.8	Pathogenic	-

Clinical significance was determined according to COSMIC (Catalogue of Somatic Mutations in Cancer) and cBioPortal databases. ACA: atypical angiofibroma, CAS: angiofibroma with sarcomatous transformation, VAF: variant allele frequency. Two cases with significant *TP53* VAF value were highlighted as bold.

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
