# Peer review of "First Glance of Molecular Profile of Atypical Cellular Angiofibroma/Cellular Angiofibroma with Sarcomatous Transformation by Next Generation Sequencing"

_diagnostics, 2020, doi:10.3390/diagnostics10010035_

Round 1

Reviewer 1 Report

It is not clear what authors mean by the sentence in the abstract: “Atypical cellular angiofibroma and cellular angiofibroma with sarcomatous transformation are recently described variants showing worrisome morphological features and strong, diffuse p16 expression.” Do they mean variants characterized in ACA and CAS?

For Table 2, authors need to indicate the transcript IDs (NM and XM prefix reference sequences). They should also provide rs numbers if available, to make table more accessible to readers.

VAF for TP53 are very high. They may be germline mutations, not somatic mutations. Authors need to discuss about that.

Bioinformatic analysis should be described in detail. E.g. Which genome build was used for alignment?

Reviewer 2 Report

Since the mutation analysis has been done on FFPE material, it is important to be aware that formalin induces DNA damage that typically results in artifactual C->T mutations present at low allele frequencies. A higher cutoff than 2% may be justified for reporting variants in this case, as low frequency mutations may be more likely to be artifacts rather than subclonal events. The exception may be if these occur at known hotspot sites.

It should also be noted that the labels used in ClinVar do not necessariliy indicate variants that are of relevance to cancer, since they are defined in the context of specific Mendelian diseases. While a variant labled as ”pathogenic” is likely to imply that is can have a consequence on protein function, this does not necessarily imply that it is pathogenic with respect to cancer. ClinVar is not really intended for assessing the consequences of somatic variation. The of ClinVar labels to indicate the clinical significance of variants with respect to tumors in this manuscript is therefore misleading. It is better to investigate the frequencies of individual mutations in different cancer cohort, for instance by comparing with TCGA data in cBioPortal.

Since it appears that matching normal samples were not used for mutation-calling, the autors should compare the listed mutations to variants present in population databases, such as dbSNP, 1000 genomes and/or ExAC to assess whether they are likely to be somatic.

For the NGS analyses, reference genome build should be described, as well as which versions of the COSMIC and ClinVar databases were used. Ideally, the sequencing data would also be submitted to an online repository.

Are there any previous literature on any of the discovered variants, besides RB1 and TP53, in similar tumor types?

In Table 1, legends explaining the lables 1+, 2+, 3+ should be provided.

Fig. 1: The legend states that A represents a usual cellular angiofibroma and B states that that the image shows an abrupt transition to a region with hypercellularity and nucler atypia. The way this is written, it is unclear whether these two sections are from the same or from different patients.

Page 3: ”Fluorescence in situ hybridization (FISH) examination showed a monoallelic deletion of RB1 gene in all 6 ACA/CAS cases, similar to the control CA samples (Fig. 2D).” – Only one case is shown in this figure. Results for the other cases should be shown to support this statement.

Page 3: ”No significant genetic variants could be demonstrated in any of the control CA cases.” – If significance here refers annotation with ClinVar, then see comment 2.

Page 3: ”p53 immunohistochemical stainings for both cases showed strong intranuclear positivity within the sarcomatous component as well (Fig. 2C)”. - Only one of the samples is shown in the figure. Both samples should to be shown to support this statement.

Page 3: ”One case of ACA with scattered atypical cells possessed hyperchromatic nuclei (Fig. 1) and the other 5 cases of CAS had two to three PLPS- and UPS-like areas (Fig. 2A).” – Fig. 2A only shows one case.

Page 3: ”Immunohistochemically, all ACA/CAS cases showed positivity for CD34, oestrogen-/progesterone-receptor and also p16 (Fig. 2B). None of the cases showed positivity for MDM2 and CDK4. Both control cases were negative for p16 by immunohistochemistry.” – Fig. 2B only shows p16 expression, and only for one sample. To support this statement, staining restults concerning the other genes and for all samples should to be included.

No evidence for statements concerning the two control samples appear to be presented anywhere in the manuscript.

Discussion: ”In summary, this is the first study performing large scale mutation profiling of ACA/CAS by NGS” – 6 samples and 67 genes analyzed by targeted sequencing on FFPE material without matched normal controls can hardly be described as a large-scale mutation profiling.

Discussion: ” It has been documented that ACA/CAS often shows either multifocal or diffuse p16 overexpression compared with either scattered or negative expression in CA indicating to additional molecular pathways crucial in ACA/CAS pathogenesis. [...] p53 deficiency may be involved in the tumorigenesis inducing p16 overexpression in selected cases.” – The discussion seems to posit that p16 overexpression is a central player driving the aberrant behaviour of these cells. Given that p16 is a tumor suppressor, it is unlikely that increased expression of p16 is the relevant endpoint of p53 deficiency. More likely, it is a side-effect of failed genome maintenence and a marker of senescence, rather than being a protein that in itself contributes to driving the proliferation of these cells.

Discussion: ”However, low frequency benign/undetermined gene variants indicated subclonal changes directing to an enhanced genomic instability in ACA/CAS.”- Given that the material is FFPE-treated, it is not certain that these are true subclonal variants. They could also be artifacts induced by formalin. In addition, it does not appear that overall genomic stability has been assessed here (one might, for instance consider profiling copy number changes with NGS or array-based methods to assess this).

Discussion: ”Following DNA isolation and successful sequencing from archived tissue samples a small set of gene variants was identified in the CAS cases but none in the CA or ACA cases, supporting the progressive nature of the former.” – Not necessarily. Given the small set of genes assessed, the other samples could easily have mutations in an entirely different set of genes that are not found in CAS and therefore follow a different evolutionary trajectory. Discussion: ” The presence of RB1 gene deletion (e.g. by FISH analysis) has been also found useful as a characteristic marker to differentiate from WDLPS, PLPS and UPS since the former has MDM2 gene amplification and the latter two possess complex karyotypes.” – Did the authors investigate MDM2 status or karyopic appearances in the CAS/ACA/CA cases (besides the RB1 locus)?

Round 2

Reviewer 1 Report

I believe that authors fully adressed the comments and the manuscript is now appropriate for publication.

Author Response

We thank you for your review.

Reviewer 2 Report

The following comments have not yet been satisfactorily resolved. Below are the original comments, the authors’ responses and follow-up replies to those responses:

Major:

2. It should also be noted that the labels used in ClinVar do not necessariliy indicate variants that are of relevance to cancer, since they are defined in the context of specific Mendelian diseases. While a variant labled as ”pathogenic” is likely to imply that is can have a consequence on protein function, this does not necessarily imply that it is pathogenic with respect to cancer. ClinVar is not really intended for assessing the consequences of somatic variation. The of ClinVar labels to indicate the clinical significance of variants with respect to tumors in this manuscript is therefore misleading. It is better to investigate the frequencies of individual mutations in different cancer cohort, for instance by comparing with TCGA data in cBioPortal.

Author response: We apologized that we are not familiar with TCGA data in cBioPortal. However, our result achieved from NGS analysis was confirmed by ClinVar and further double-checked by COSMIC database to ensure such genetic aberrations have clinical significance in tumorgenesis.

Follow-up comment: Presence in COSMIC does not prove that the variants have clinical relevance in tumorigenesis. Many genes have at some point been found mutated in cancer and the vast majority are passenger events without consequence. The frequencies at which these mutations occur are what is relevant and this piece of information needs to be assessed in order to state that they are relevant to cancer. And, as mentioned, ClinVar labels are not necessarily relevant to cancer and the use of these labels as a basis for discussing the clinical significance of the variants is misleading.

3. Since it appears that matching normal samples were not used for mutation-calling, the autors should compare the listed mutations to variants present in population databases, such as dbSNP, 1000 genomes and/or ExAC to assess whether they are likely to be somatic.

Author response: We agree with reviewer that we did not use normal matching samples for mutation calling. Nevertheless, we used gnomAD population database to compare the significance of each gene alterations which is included in our Archer NGS analysis system.

Follow-up comment: gnomAD is an acceptable database to compare with, but the criteria that were used to exclude variants present in gnomAD need to be described. A certain minor allele frequency? Binary presence in the database?

8-13 and 18. Follow-up comment: The authors’ have not provided the IHC stainings requested to support their statements. The brevity of a communications-style report is not an excuse if supplementary data can be included. To cite the “Instructions of authors” document: “In order to maintain the integrity, transparency and reproducibility of research records, authors must make their experimental and research data openly available either by depositing into data repositories or by publishing the data and files as supplementary information in this journal.”

Minor

17. Discussion: ”Following DNA isolation and successful sequencing from archived tissue samples a small set of gene variants was identified in the CAS cases but none in the CA or ACA cases, supporting the progressive nature of the former.” – Not necessarily. Given the small set of genes assessed, the other samples could easily have mutations in an entirely different set of genes that are not found in CAS and therefore follow a different evolutionary trajectory.

Author response: We have modified our manuscript to address this point.

Follow-up comment: The revised sentence reads as follows: “Following DNA isolation and successful sequencing from archived tissue samples a small set of gene variants was identified in the CAS cases but none in the CA or ACA cases, might support the higher mitotic activity and larger size of the former.” The way this is written is not entirely grammatically correct. A better way to phrase this would be: “Following DNA isolation and successful sequencing from archived tissue samples a small set of gene variants was identified in the CAS cases but none in the CA or ACA cases, which might explain the higher mitotic activity and larger size of the former.”

In addition to the above, a certain amount of language improvement is desired.

Round 3

Reviewer 2 Report

Related to the response to comment 2: It appears correct that the majority of these mutations are not present in any of the samples profiled in cBioPortal, suggesting that their relevance in cancer may be low. At least some of them are represented, however: TP53 M1 (1 sample), KIT V50M (4 samples), STK11 F354L (5 samples), TP53 R110C (8 samples), TP53 Y163C (79 samples), JAK3 T714M (1 sample), NRAS A146T (2 samples). Among these, at least the TP53 Y163C mutation can be considered a significant mutational hotspot in cancer.

Again, the main concern here was that, in the text, ClinVar labels are as the sole piece of evidence to draw conclusions about clinical significance in cancer. The very least that can be done to address this point is to state the limitations of those labels (that they apply mainly to Mendelian diseases) in the context of cancer in the Discussion section, if no other source of evidence is used to back up those claims about significance.

To enable further interpretation of these variants, as pointed out by reviewer 1, one also needs to know which transcripts they occur in. While transcript IDs are listed in Table 1 for some of the genes not all of them have such IDs indicated. Why not?

Related to the response to comments 8-13 and 18: The added supplementary figures would resolve this issue, but they raise a couple of questions: 1) The FISH images are somewhat difficult to interpret: It is stated that all samples have RB1 deletion. If a single red dot implies RB1 loss, then the FISH images in the supplementary material in fact shows two red dots for samples 1-4, while samples 5-6 instead shows 1 dot. How does this indicate RB1 loss in samples 1-4? 2) Are the MDM2 and CDK4 stainings for samples 3-4 iterative stainings or duplicated figures (the resolution is too low to tell whether they are different)? 3) Would it be possible to include the control samples here as well?
